# Spatial Characteristics of Urban Green Spaces and Human Health: An Exploratory Analysis of Canonical Correlation

**DOI:** 10.3390/ijerph17093227

**Published:** 2020-05-06

**Authors:** Chia-Tsung Yeh, Ya-Yun Cheng, Tsai-Yun Liu

**Affiliations:** 1Graduate Institute of Urban Planning, National Taipei University, No. 151, University Rd., Sanxia Dist., New Taipei City 237303, Taiwan; 2Urban and Rural Development Bureau, New Taipei City Government, No. 161, Sec. 1, Zhongshan Rd., Banqiao Dist., New Taipei City 22001, Taiwan; 0910129@gmail.com (Y.-Y.C.); AM5180@ms.ntpc.gov.tw (T.-Y.L.)

**Keywords:** urban green spaces, spatial characteristics, human health, canonical correlation analysis

## Abstract

In highly urbanized areas, urban green spaces (UGSs) are important natural and cultural entities. Previous studies have shown some evidence of positive relationships between UGSs and human health. Most of these studies relied on self-reported health data and often used institutional quantitative measures of UGSs instead of the spatial characteristics of UGSs. This study analyzed the relationships between the spatial characteristics of UGSs and morbidity of diseases, which were considered variables of human health in the Taipei Metro. The Longitudinal Health Insurance Database of Taiwan was applied as the source of morbidity of diseases. A canonical correlation analysis was performed by using the six variables of spatial characteristics of UGSs as predictors and three variables of morbidity as criterion variables to evaluate the multivariate shared relationships between the two variable sets. The results found a strong canonical correlation between the spatial characteristics of UGSs and human health. Furthermore, the results revealed that living in districts with a higher area percentage of green spaces and denser vegetation cover, as well as exposure to more aggregative and irregular-shape green spaces, can reduce the morbidity of diseases.

## 1. Introduction

Green space can be defined as lands that consist predominantly of unsealed, pervious, soft surfaces, such as soil, grass, shrubs, and trees, with few structures present [1]. In highly urbanized areas sealed by artificial constructions, green spaces are important natural and cultural entities for urban residents. Green spaces in urban areas provide important ecosystem services, such as air quality improvement, atmospheric carbon dioxide reduction, and recreational and cultural values [2,3,4]. However, urbanization, environmental degradation, and lifestyle changes are diminishing opportunities for human contact with nature [5].

Amenities in green spaces encourage urban residents to increase physical activities, to pursue more leisure and recreation, and to have contact with the natural environment. Increasing evidence indicates that the presence of green spaces can enhance the quality of life [6,7]. For example, easy access to green spaces increases physical activity [5,8,9] and promotes physical and mental health of urban residents [6,10,11,12,13]. More greenness in urban residential areas is associated with more physical activity and lower risk of disease mortality [14]. Moreover, urban green spaces (UGSs) can facilitate social interactions, social networking, social support, social inclusion, and improve the mental health of residents [15,16,17].

More people live in cities now than at any other point in history. Urban growth is presenting numerous challenges for the maintenance of urban green spaces, and, consequently, for human health. An emerging body of studies has linked exposures to UGSs with improving human health, through increasing physical activity, reducing levels of noise and pollution, relieving physiological and psychological stress, and enhancing social contacts [18,19,20,21,22,23,24,25]. Epidemiological studies have focused on the association between UGSs and the risks of specific diseases, such as obesity (e.g., [26]), cardiovascular disease (e.g., [21,27,28,29]), and type 2 diabetes (e.g., [30,31]). However, few studies have simultaneously investigated the relationships between the spatial characteristics of UGSs and the risks of two or more diseases [11].

Previous studies heavily relied on self-reported data to investigate the association between exposures to UGSs and health benefits to participants (e.g., [20,24,25,32,33]). Questionnaire surveys and interviews are frequently used to obtain the self-reported data [17,34,35]. The criterion variables used in the analyses of associations were self-reported health indicators, e.g., number of symptoms experienced in the last 14 days, perceived general health, and score on the general health questionnaire [19,24]. These studies had relatively small study population sizes or were cross-sectional in design, limiting the generalizability of the results [29]. However, rare studies conducted their studies based on the large-scale datasets of human health insurance and analyzed the association between UGSs and the risk of disease in metropolitan areas [29].

Moreover, previous studies focused on the relationships between UGSs and human health usually used institutional measures of UGSs (e.g., area of green spaces per capita) instead of quantitative spatial characteristics of UGSs (e.g., composition, spatial configuration, and spatiotemporal patterns of UGSs). Green spaces with a complicated shape could bring people closer to nature, improve their physical and mental health [36]. In addition, an elongate park could be accessible by more residents, and a connected green network has higher amenity value than smaller and fragmented ones [37]. The fragmentation and separation of green space contributed to increased mortality of cardiovascular disease through air pollution and temperature [28]. The quantitative indicators, which refer to morphologies and vegetation cover, and spatial distribution, are considered as three main dimensions to assess UGSs effectively [38]. Quantifying the spatial characteristics of UGSs by using a geographic information system (GIS) and landscape metrics provides a more systematic and precise way to analyze the spatial characteristics of UGSs.

The variables of spatial characteristics of UGSs and human health possibly have multiple causes and effects between them. Determining outcomes based on a correlation analysis between characteristics of UGSs and morbidity of specific diseases may be insufficient to indicate the general health benefit of UGSs. Moreover, morbidities of different diseases may have disparate relationships with the characteristics of UGSs. In studies that only consider one criterion measure of human health, a multiple regression analysis can be conducted. However, this study tended to examine the relationships between two variable sets, i.e., the predictor variables (spatial characteristics of UGSs) and criterion variables (morbidities of diseases). Therefore, canonical correlation analysis (CCA) was applied to determine which variables are more or less influential in the model. The National Health Insurance System of Taiwan provides a high-quality dataset of outpatient treatments and hospitalizations with an accurate disease classification scheme. This dataset was the base to derive the morbidity of diseases, i.e., cardiovascular diseases, mental disorders, and respiratory diseases, as observed variables of human health in this study.

The following method section introduces the research methods, the study area, data sets required, and techniques for quantifying spatial characteristics of UGSs. Additionally, this section briefly introduces canonical correlation analysis, the multivariate statistical method used in this study. The section of results reveals the key variables of UGS on human health and the roles of variables of characteristics of UGS based on an exploratory statistical analysis. The discussion section compares the key findings in this study with current knowledge reported in recent studies. The last section concludes with the significant correlations between UGSs and human health and the advantages of using the data derived from the national health insurance system.

## 2. Methods

### 2.1. Study Area

In this study, the largest metropolitan area in Taiwan, Taipei Metro Area, was selected for the analysis of the relationships between spatial patterns of green spaces and human health. Figure 1 shows that Taipei Metro Area has three cities, i.e., Taipei City, New Taipei City, and Keelung City, and 48 districts and has a total area of 2457 km^2^, with a population of 7.04 million at the end of 2017, representing 30% of the total population of Taiwan [39]. The mean annual temperature of Taipei Metro Area is 23 °C, with the variation between the lowest monthly average temperature of 16 °C in January and the highest monthly average temperature of 29.6 °C in July. The annual rainfall varies by terrain. The rainfall approximates 2400 mm in plain areas and is over 4500 mm in mountain areas (The information of rainfall and temperature data about the Taipei Metro Area was collected from [40,41,42]) This study analyzed UGSs in 37 districts that had a population larger than 25,000 and population density above 500 pop/km^2^. The remote mountain districts and rural districts with lower population or population density were excluded.

### 2.2. Spatial Characteristics of UGSs

The digital map of UGSs of the study area was derived from the Taiwan land use investigation maps for 2015 produced by the National Land Surveying and Mapping Center, Ministry of the Interior. UGSs were selected by the attributes of land use/land cover and reclassified into five types: farmland, forest, wetland, grassland, and park and open spaces. This study also selected residential land use as the focus area to assess the spatial accessibility to UGSs. The ArcGIS tool was applied to convert the map from vector format to raster format with a pixel size of 30 m × 30 m for metric analysis using the tool for measuring landscape metrics, FRAGSTATS 4.2. Satellite data of SPOT 5 on 8 May 2014 and 5 February 2012 were also used for analyzing the normalized difference vegetation index (NDVI) of UGSs by using ERDAS IMAGINE.

A large set of metrics for analyzing landscape composition and configuration has been developed, modified, and tested in the past three decades [42,43]. In this study, we targeted the landscape metrics of UGSs regarding human health based on the literature review. The proportion of green spaces (PLAND) was calculated to assess the availability of UGSs in the Taipei Metro area. This study also selected the shape of UGS (SHAPE_MN), mean distance of UGS patches to the nearest residential patch (ENN_MN), and continuity of UGS patches (COHESION) as the indicators of the spatial pattern of UGSs. All these landscape metrics were implemented by using the statistical package FRAGSTATS 4.2 [44] (See Table 1).

The quality of UGSs was measured by using NDVI to assess the greenness of vegetation in the Taipei Metro Area. The index of NDVI correlates with biophysical properties of the vegetation canopy, such as leaf area index, fractional vegetation cover, vegetation condition, and biomass [46]. NDVI is often used as a metric to assess exposure to surrounding greenness. Previous studies also showed that a higher NDVI value is related to less depression [47] and better mental health [48]. Additionally, the analysis of Network Distance of ArcGIS was used to calculate the shortest network distance from the residential area to UGS as the index of accessibility.

The landscape spatial characteristics of UGSs were analyzed for 37 districts of Taipei Metro Area. The results of the spatial analysis were used to examine the correlation between spatial characteristics of UGSs and morbidities of diseases in the study area.

### 2.3. National Health Insurance Research Database

The database of Taiwan’s National Health Insurance program contains registration files and original claim data for reimbursement. The National Health Insurance Administration, Ministry of Health and Welfare, Taiwan, derives large computerized databases from this program. The databases are maintained by the National Health Research Institutes, Taiwan, and are provided to scientists for research purposes. The data files from the National Health Insurance program are de-identified by scrambling the identification codes of both patients and medical facilities and sent to the National Health Research Institutes to form the original files of the National Health Insurance Research Database (NHIRD). This study applied the Longitudinal Health Insurance Database (LHID), which is a data subset of the NHIRD and contains all original claim data for 1,000,000 beneficiaries randomly sampled from the Registry for Beneficiaries (ID) of the NHIRD. All the registration files and original claim data of these 1,000,000 individuals were collected and constituted the LHID [49].

In Taiwan, according to the Personal Information Protection Act, residential locations of patients are confidential. Dataset of outpatients reveals only the spatial locations of the hospital. To consider the influences of cross-district outpatients, we combined the outpatient ID of the common cold (ICD9 460) and influenza (ICD9 487) with the outpatient ID of cardiovascular disease, mental disorders, and chronic respiratory disease. The locations of medical institutions for common cold and influenza are determined as the living places of outpatients based on the combined data.

This study applied the ambulatory care expenditures by visits (CD) of original claim data as the source of morbidity of diseases. The CD data contains the date of ambulatory care, ICD-9-CM codes, and ID of medical facilities (HOSP_ID). By combining ID of medical facilities (HOSP_ID) and registry for contracted medical facilities (HOSB) which includes area code of medical facilities (AREA_NO_H), the number of outpatients can be summed up by district level. We included all registered outpatient database between 1 January 2010 and 31 December 2012 and selected the database of outpatients for the three types of disease related to exposure of green space, i.e., cardiovascular diseases (ICD9 390–459), mental disorders (ICD9 290–319), and chronic respiratory diseases (ICD9 470–478, 490–519).

### 2.4. Canonical Correlation Analysis

In 1936, Hotelling developed canonical correlation analysis (CCA) to examine the correlation between a synthetic criterion variable and a synthetic predictor variable that are weighted based on the relationships between the variables within the sets [50]. Instead of analyzing a single dataset, however, the goal of CCA is to analyze the relationship between a pair of datasets [51]. Each dataset can contain several variables, and CCA calculates a linear combination for each set, called a canonical variable, such that any correlation between two canonical variables is maximized [52]. These correlations are called canonical correlations, and the linear combinations are canonical variates. The fundamental principle behind CCA is to create a number of canonical variates, each consisting of a linear combination of one set of variables (*X_i_*) in the following equation:$$U_{i}=a_{i1}X_{1}+a_{i2}X_{2}+\cdots +a_{iP}X_{P}$$

Additionally, a linear combination of the other set of variables (*Y_i_*) has the following equation:$$V_{i}=b_{i1}Y_{1}+b_{i2}Y_{2}+\cdots +b_{iq}Y_{q}$$

The goal of CCA is to determine the coefficients, or canonical weights (*a_ij_* and *b_ij_*), that maximize the correlation between canonical variates *U**_i_* and *V**_i_*. The first canonical correlation, Corr. (*U*_1_, *V*_1_), is the strongest possible correlation between a linear combination of variables in the exposure set and a linear combination of variables in the outcome set. Other pairs of maximally correlated linear combinations are chosen in turn and are orthogonal to those previously identified. The process of constructing canonical variates continues until the number of pairs of canonical variates equals the number of variables in the smaller of the two sets.

Because CCA assesses the existence of correlations between two sets of variables, this exploratory statistical method is useful for analyzing relationships between the combinations of urban form and different variables of human health. Although CCA is widely used in social science research [53], the method is rarely used in studies on UGSs [54]. This study applied CCA to examine multivariate relationships between spatial characteristics of UGSs and human health. The proposed canonical correlation model is shown in Figure 2. This study attempted to quantify the independent relationships of spatial characteristics of UGSs and levels of morbidity of three diseases. Table 2 shows the descriptive statistics of the studied variables. It provides information about the number of samples, minimum, maximum, mean, and standard deviation of the studied variables.

A canonical correlation analysis was performed with IBM SPSS Statistics for Windows, Version 22.0 (IBM Corp. released 2013, Armonk, NY, USA). Six variables of spatial characteristics of UGSs were used as predictors of three observed variables of morbidity to evaluate the multivariate shared relationship between these two variable sets. Before the analysis results show the fitness of the model, the Pearson correlation matrix describes the associations between the variables of the spatial characteristics of UGSs. It also detects the possibility of multicollinearity of studied variables. Table 3 shows that all the correlations (r) < 0.765 (in absolute value). The general rule of thumb is that if the correlation (r) > 0.8, then severe multicollinearity may be present. Therefore, it is not necessary to remove the redundant variables.

This study uses a multivariate test of significance of CCA (i.e., Pillai’s, Hotelling’s, Wilk’s, and Roy’s multivariate criteria) to show the general fit of the model. Furthermore, the canonical correlation coefficients and the eigenvalues are used to explain the fitness of the canonical roots of the model. Based on the tests, the significance of each of the roots in the dimension reduction analysis of SPSS, the possible significant roots are found. Since our model contains three disease morbidities and six indicators of spatial characteristics of UGSs, SPSS extracts three canonical roots or dimensions. In the dimension reduction analysis, the first test of significance tests all three canonical roots of significance; the second test excludes the first root and tests roots two to three; the last test tests root three by itself.

## 3. Results

### 3.1. Spatial Pattern of the Characteristics of UGSs and the Morbidity of Diseases

We analyzed the spatial characteristics of UGSs of the districts in the study area based on the land use investigation map and the FRAGSTATS 4.2 software. Figure 3a shows that the districts with a low percentage of green spaces roughly clustered in the center of the Taipei Metro Area. The level of NDVI of the districts (Figure 3b) shows that UGS quality was lower in the urban central districts than that in peripheral districts. The level SHAPE_MN of the districts (Figure 3c) similarly shows that the shape of UGSs in the urban central districts was more regular compared to peripheral districts. In highly urbanized districts, the distances between green spaces (ENN_MN) were longer than those in the lightly urbanized out-ring districts (Figure 3d). Conversely, the continuity of green spaces (COHESION) was higher in peripheral districts compared to central districts (Figure 3e). The Network Distances from residential areas to green spaces were longer in the central districts (Figure 3f). Generally, availability, quality, and accessibility of UGS were higher in southern and northern west parts of the Taipei Metro Area. Inversely, availability, quality, and accessibility of UGSs of the districts in the Taipei basin area and Keelung city were poorer. The connectivity of UGS was lower in highly urbanized districts.

Morbidities of cardiovascular disease, mental disorder, and chronic respiratory disease were high, and the quality of green spaces was low in the central districts of the Taipei Basin. Conversely, western and southern districts had low morbidity and high quality and connectivity of green spaces. The spatial patterns of morbidity of cardiovascular disease, mental disorder, and chronic respiratory disease (Figure 4) were roughly opposite to the quality, greenness, and connectivity of green spaces in our study area.

### 3.2. Relationships Between Spatial Characteristics of UGSs and Morbidity

The output of CCA shows the general fit of the model reporting Pillai’s, Helling’s, Wilk’s and Roy’s multivariate criteria. The commonly used test is Wilk’s, but we found that all of these tests were significant, with *p* < 0.001 (see Table 4).

Table 5 reports the canonical correlation coefficients and the eigenvalues of the canonical roots. The CCA analysis yielded three canonical functions with squared canonical correlation (Sq Cor.) of 0.62941, 0.40733, and 0.09710 for each root. The first pair of variates, a linear combination of spatial characteristics of UGSs and a linear combination of the variables of morbidities of diseases, had a canonical correlation coefficient of 0.79336 and an eigenvalue of 1.698. The analytical results indicate that the spatial characteristics of UGSs and human health are correlated. The first root was created to maximize the canonical correlation (Pearson *r*) between the two synthetic variables. The second pair of canonical variates was constructed out of the residuals of the first pair to maximize the correlation between them. The first and second roots had explained variances of 68.12% and 27.57%. The third root had a low explained variance of 4.31% (see Table 5).

SPSS extracted three canonical roots or dimensions for our model. The dimension reduction analysis tested the significance of each of the roots (see Table 6). The first test of significance showed the full model across root 1 to 3 was significant (Wilk’s λ = 0.19831, F = 3.41680, *p* < 0.01). The second test excluded the first root and tests roots 2 to 3 and showed it was also significant (Wilk’s λ = 0.53512, F = 2.21872, *p* = 0.036 < 0.05). The root 3 itself was not significant (Wilk’s λ = 0.90290, F = 0.80659, *p* = 0.531) (see Table 6).

Next, we turned to the research question of which variables of the spatial characteristic of UGSs contribute to this relationship between the variable sets across the two functions. Table 7 shows the statistics of the canonical solution of Function 1 and Function 2 for the relationships between spatial characteristics of UGSs and morbidities of diseases. The table presents the structure coefficients (r_s_), the squared structure coefficients (r_s_^2^), and cross-structure coefficients (r_c_) between a set of variables and the synthetic variable created by another set of variables.

In Function 1, the first pair of canonical variates groups the variables in the way that the correlation between them is maximized. The structure coefficients (r_s_) of PLAND, NDVI, SHAPE_MN, ENN_MN, and Network Distance indicated that they were the primary contributors (with r_s_ = −0.53151, −0.60826, −0.48613, 0.53458, and 0.88403) to the synthetic variate of characteristics of green spaces. The cross-structure coefficients (r_c_) of PLAND, NDVI, and SHAPE_MN showed that they were negatively related to the synthetic variate of morbidity of diseases. Inversely, the cross-structure coefficients (r_c_) for ENN_MN and Network Distance show that they were positively related to the synthetic variate of morbidity of diseases. The other side of Function 1, the structure coefficients (r_s_) of criterion variables indicate that cardiovascular diseases, mental disorders, and respiratory diseases were positively correlated with the synthetic variate of morbidity (with very high *r_s_* = 0.98556, 0.98352, and 0.99449). The cross-structure coefficients (*r_c_*) of the three criterion variables, all of which were higher than 0.7, revealed that morbidity of cardiovascular diseases, mental disorders, and respiratory diseases all had positive correlations with the synthetic variate of spatial characteristics of UGSs.

In the Function 2, the values of structure coefficients (r_s_) indicated that the variables of PLAND, NDVI, and COHESION were the primary contributors to the synthetic variate of predictors. The cross-structure coefficients (r_c_) of PLAND, NDVI, and COHESION showed that they were positively related to the synthetic variate of disease morbidities with moderate r_c_ = 0.34186, 0.33197, and 0.47943. On the other side of Function 2, the structure coefficients (r_c_) of criterion variables indicated that cardiovascular diseases, mental disorders, and respiratory diseases had very low correlation coefficients with the synthetic variate of morbidity of diseases (with very low *r_s_* < 0.2). The cross-structure coefficients (*r_c_*) of all the three criterion variables were lower than 0.12, revealing that morbidity of cardiovascular diseases, mental disorders, and respiratory diseases all had very low correlations with the synthetic variate of spatial characteristics of UGSs.

## 4. Discussion

In this study, we correlated the district-level spatial characteristics of UGSs with the morbidities of specific diseases. The results of CCA analysis showed a significant correlation between spatial characteristics of UGSs and human health in the Taipei Metro Area. The cross-structure coefficients of predictor variables revealed that the morbidity of diseases was decreased by residence in the districts with higher area percentage, denser vegetable cover, more aggregative urban greens, and irregular-shape green spaces. In comparison with the variables of spatial characteristics of UGSs, accessibility (measured by Network Distance) was the major contributor to the relationship between spatial characteristics and human health. The secondary contributors were the variables of quality of UGSs (measured by NDVI) and spatial pattern of UGSs (measured by SHAPE_MN, ENN_MN, and COHESION). The availability of UGSs (measured by PLAND) was less influential in the relationship between spatial characteristics of UGSs and human health.

The results of correlation analyses validated the evidences of health benefits of exposure to green spaces that have been found by previous studies (e.g., [21,35,55,56,57]). This study also revealed the key variables of spatial characteristics of UGSs in the relationship between UGSs and human health. The analytical results showed that the variables of quality (measured by NDVI) and accessibility played key roles in influencing the morbidities of diseases in the Taipei Metro Area. These findings confirm the evidence found by [9,12,34] in their study areas. In this study, the strategy of using a database of national health insurance and conducting district-level analyses is similar to the newly published study of [29]. However, the previous study usually concerned only with a single disease, i.e., cardiovascular disease. Our study reveals the correlations between the spatial characteristics of UGSs and the morbidity of three diseases.

A major limitation of our study is that the data of the National Health Insurance Research did not reveal the real addresses of the patients but only the districts where the patients were insured and outpatient clinics located. However, the locations of medical institutions they visited sometimes may be different from that they lived. To reduce the effects of cross-district outpatients, we cross-matched the address of the medical institution which the patients visited for the common cold and flu. This match was based on the assumption that urban residents tend to visit medical institutions near to their dwelling places to get timely treatment for a light ailment, such as cold and flu.

The database of residential health used in this study is different from self-reported data used by the previous studies that investigate the association between exposures to UGSs and health status of interviewees (e.g., [25,27,32,33,58]) We do not know their real contact time to green spaces and individual behavior of urban residents based on the database of the National Health Insurance. However, this approach is better than that using self-reported data in the case of dealing with the relationship between UGSs and health status for whole residents in a big metropolitan area.

## 5. Conclusions

Many previous studies focusing on the health benefits of urban green spaces relied on self-reported health and socio-economic data from a relatively small number of respondents. Using the database of the National Health Insurance Research of Taiwan and applying the statistical method of canonical correlation analysis distinguish this study from previous ones. This study shows the detailed relationships between spatial characteristics of UGSs and three diseases that humans suffer from based on the data of morbidities of three diseases. The analytical results revealed a significant positive correlation between the spatial characteristics of UGSs and human health. The results also showed that the districts with lower availability, quality, accessibility to green spaces had a high level of morbidity of cardiovascular diseases, mental disorders, and respiratory diseases. Conversely, districts with high aggregation, availability, quality, and accessibility to green spaces apparently had a low level of morbidity.

Since previous studies of UGSs and human health have focused on the risks of a specific disease, this study investigated the relationships between the spatial characteristics of UGSs and levels of general human health. Instead of relying on the morbidity of a specific disease, this study simultaneously investigated the relationships between the morbidities of three diseases and the spatial characteristics of UGSs. The analytical results reveal that human health generally has a strong correlation with the spatial characteristics of urban green spaces. The results also reveal that better biophysical property of vegetation canopy in urban areas is very important to improve human health. We emphasize not only the importance of the availability of UGSs for human health but the contributions of quality, accessibility, and spatial patterns of UGSs to human health. This study revealed that Network Distance and NDVI of UGSs are the main contributors to human health. Respiratory disease is shown by our results as the most influenced disease by spatial characteristics of urban green spaces. In addition to increasing the number of UGSs, improving the quality, accessibility, and spatial patterns of UGSs can reduce the morbidities of diseases and promote better urban human health. The findings of this study can be useful for maintaining precious green spaces in cities and for urban planning to improve green infrastructure in urban areas.

## Figures and Tables

**Figure 1 ijerph-17-03227-f001:**
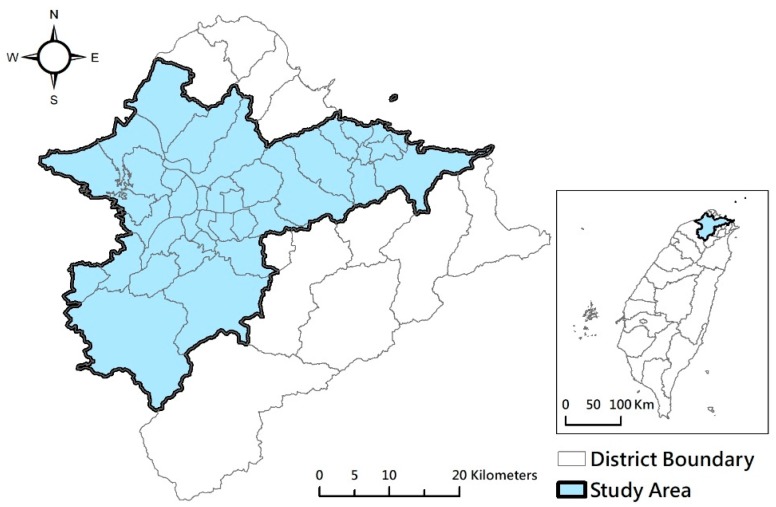
Study area.

**Figure 2 ijerph-17-03227-f002:**
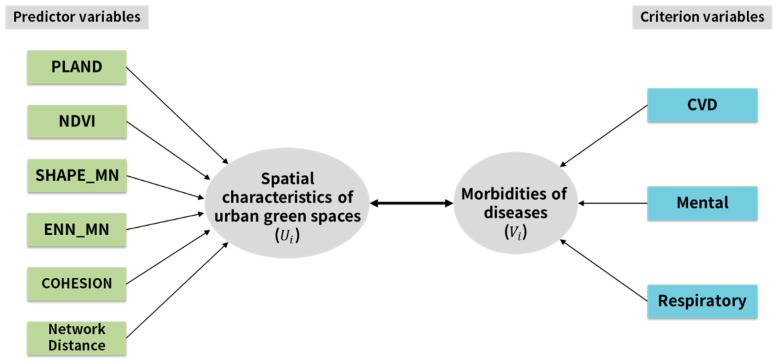
The canonical correlation model that was used in our study to assess the relationship of the predictor variables (spatial characteristics of urban green spaces (UGSs)) with criterion variables (morbidities of CVD (cardiovascular diseases), mental disorders, and respiratory diseases).

**Figure 3 ijerph-17-03227-f003:**
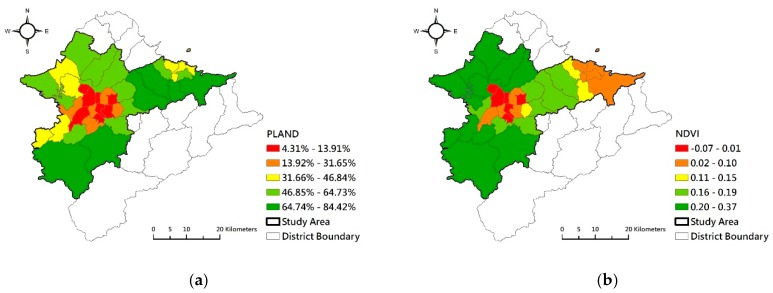
Landscape spatial patterns and distribution of urban green spaces. (**a**) proportion of green spaces (PLAND); (**b**) normalized difference vegetation index (NDVI); (**c**) shape of UGS (SHAPE_MN); (**d**) mean distance of UGS patches to the nearest residential patch (ENN_MN); (**e**) continuity of UGS patches (COHESION); (**f**) Network Distance.

**Figure 4 ijerph-17-03227-f004:**
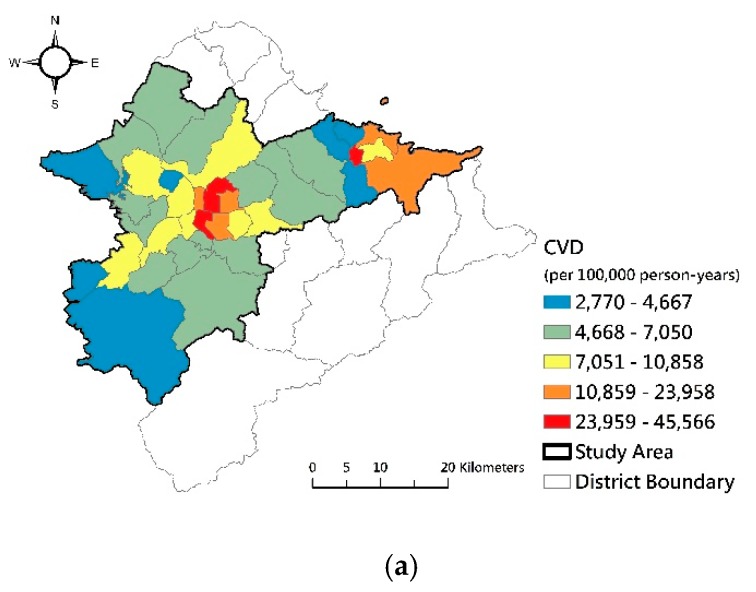
Levels of the morbidity of the three types of diseases. (**a**) Cardiovascular diseases; (**b**) Mental disorders; (**c**) Respiratory diseases.

**Table 1 ijerph-17-03227-t001:** Landscape spatial characteristics of urban green spaces (UGSs).

Four Aspects	Indicators	Description
Availability	PLAND	Percentage of study area comprising all patches of UGSs
Spatial Pattern	SHAPE_MN	Shape complexity of UGSs calculated as patch perimeter (m) of urban green spaces divided by the theoretically minimum perimeter of patch area (m)
	ENN_MN	Mean distance (m) of the patches of UGSs to the nearest neighboring patch based on shortest edge-to-edge distance from cell center to cell center
	COHESION	Connectivity of the patches of UGSs by computing the patch area and perimeter
Quality	NDVI	Indication of exposure and condition of green vegetation in UGSs [27]
Accessibility	Network Distance	Shortest distance between the residential land use and UGSs along a road network [45]

PLAND: Percentage of landscape; SHAPE_MN: Mean patch shape index; ENN_MN: Mean Euclidean nearest-neighbor distance; COHESION: Patch cohesion index; NDVI: Normalized difference vegetation index. Source: [44]. FRAGSTATS v4.2: Spatial Pattern Analysis Program for Categorical Maps. Computer software program produced by the author at the University of Massachusetts, Amherst, MA, USA.

**Table 2 ijerph-17-03227-t002:** Descriptive statistics of the studied variables.

Diseases	N	Minimum	Maximum	Mean	Std. Deviation
CVD (per 100,000 person-year)	37	2770	45,566	11,090.81	9870.864
Mental (per 100,000 person-year)	37	821	14,735	3683.97	3214.976
Respiratory (per 100,000 person-year)	37	1185	21,270	5140.27	4694.280
PLAND (%)	37	0.0431	0.8442	0.427227	0.2415632
SHAPE_MN (none)	37	1.26	2.15	1.6136	0.16357
ENN_MN (m)	37	65.65	195.36	102.4700	35.77035
NDVI (none)	37	−0.07	0.37	0.1578	0.12079
Network distance (m)	37	207.79	888.71	443.2970	162.14926
COHESION (none)	37	76.84	99.96	97.2686	4.67171
Valid N (listwise)	37				

**Table 3 ijerph-17-03227-t003:** Correlation analysis of the variables of spatial characteristics of UGSs.

Spatial Characteristics of UGSs.	PLAND	SHAPE_MN	ENN_MN	NDVI	Network Distance	COHESION
PLAND	1	0.234	−0.633	0.674	−0.756	0.708
SHAPE_MN	0.234	1	−0.241	0.162	−0.160	0.446
ENN_MN	−0.633	−0.241	1	−0.408	0.526	−0.427
NDVI	0.674	0.162	−0.408	1	−0.712	0.671
Network Distance	−0.756	−0.160	0.526	−0.712	1	−0.396
COHESION	0.708	0.446	–0.427	0.671	−0.396	1

**Table 4 ijerph-17-03227-t004:** Multivariate tests of significance.

Test Name	Value	Approx. F	Hypoth. DF	Error DF	Sig. of F
Pillais	1.13385	3.03793	18.00	90.00	0.000
Hotellings	2.49326	3.69372	18.00	80.00	0.000
Wilks	0.19831	3.41680	18.00	79.68	0.000
Roys	0.62941				

**Table 5 ijerph-17-03227-t005:** Eigenvalues and canonical correlations.

Root No.	Eigenvalue	%	Cum. %	Canon Cor.	Sq. Cor.
1	1.69843	68.12088	68.12088	0.79336	0.62941
2	0.68728	27.56565	95.68653	0.63823	0.40733
3	0.10755	4.31347	100.00000	0.31161	0.09710

**Table 6 ijerph-17-03227-t006:** Dimension reduction analysis.

Roots	Wilks L.	F	Hypoth. DF	Error DF	Sig. of F
1 TO 3	0.19831	3.41680	18.00	79.68	0.000
2 TO 3	0.53512	2.12872	10.00	58.00	0.036
3 TO 3	0.90290	0.80659	4.00	30.00	0.531

**Table 7 ijerph-17-03227-t007:** Canonical correlation solution.

**Predictor Variables**	**Function 1**			**Function 2**		
	***r_s_***	***r_s_^2^***	***r_c_***	***r_s_***	***r_s_^2^***	***r_c_***
PLAND	−0.53151	0.28250	−0.42168	0.53563	0.28690	0.34186
NDVI	−0.60826	0.36998	−0.48257	0.52014	0.27054	0.33197
SHAPE_MN	−0.48613	0.23632	−0.38568	−0.17472	0.03052	−0.11151
ENN_MN	0.53458	0.28578	0.42411	−0.03399	0.00115	−0.02169
COHESION	−0.34953	0.12217	−0.27730	0.75119	0.56428	0.47943
Network Distance	0.88403	0.78150	0.70135	−0.28024	0.07853	−0.17886
**Criterion Variables**	**Function 1**			**Function 2**		
	***r_s_***	***r_s_^2^***	***r_c_***	***r_s_***	***r_s_^2^***	***r_c_***
Cardiovascular diseases	0.98556	0.97139	0.78190	−0.11529	0.01344	−0.07358
Mental Disorders	0.98352	0.96731	0.78029	−0.17786	0.03163	−0.11351
Respiratory Diseases	0.99449	0.98901	0.78899	−0.06647	0.00442	−0.04242

*r_s_* = structure coefficient; *r_s_^2^* = squared structure coefficient; *r_c_* = cross-structure coefficient. For emphasis, absolute value of structure coefficients and cross-structure coefficients above 0.45 are underlined.

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
