# Peer review of "Spatial Characteristics of Urban Green Spaces and Human Health: An Exploratory Analysis of Canonical Correlation"

_ijerph, 2020, doi:10.3390/ijerph17093227_

Round 1
Reviewer 1 Report
Review ijerph-761558
The paper is well written and the question it tries to address is relevant. The original contribution of the study, as the authors argumented and I agree with them, is the use of large public datasets and spatial characteristics. This study tried to validade the results of previous studies about the same subject/question and I believe it managed to do so.
I have some suggestions though about the writing style and about the methods.
Writing style:
Lines:
14-15: “... and morbidity of diseases which considered as variables of human health in Taipei Metro.”
The sentence lacks something. Maybe the authors wanted to say: “... and morbidity of diseases which were considered as variables of human health in Taipei Metro.”?
35-36: Where it is written “to pursue more leisure and recreation, to contact with natural environment.”
Maybe rewrite to “to pursue more leisure and recreation, and to contact with natural environment.”
51-53: Where it is written: “cardiovascular disease (e.g. Pereira et al., 2012; Tamosiunas et al., 2014; Shen and Lung, 2016; Seo et al., 2019), type 2 diabetes (e.g. Astell-Burt et al., 2014; Bodicoat et al., 2014).”
Maybe rewrite to “cardiovascular disease (e.g. Pereira et al., 2012; Tamosiunas et al., 2014; Shen and Lung, 2016; Seo et al., 2019), and type 2 diabetes (e.g. Astell-Burt et al., 2014; Bodicoat et al., 2014).”
64-65: Where it is written: “There is only a new study extracted study population more than 300 thousand participants...”
Maybe rewrite the sentence because it is confusing.
88-89: Where it is written: “Therefore, canonical correlation analysis (CCA) was applied to determine what variables are more or less influential in the model.”
Maybe rewrite to: “Therefore, canonical correlation analysis (CCA) was applied to determine which variables are more or less influential in the model.”
111: The unit of measurement of rainfall is in meters? 2,400m in plain areas? Maybe converto to mm or in.
155-156: Where it is written: “which is a data subset of NHIRD, contains all original claim data for 1,000,000 beneficiaries randomly sampled...”
Maybe rewrite to: “which is a data subset of NHIRD, and contains all original claim data for 1,000,000 beneficiaries randomly sampled...”.
Methods:
Section 2.1: This section lacks a reference for all the geographical information the authors provided about the Taipei Metro Area like rainfall, temperature, etc.
Section 2.2: Please, add a small paragraph explaining how you presented the results of the spatial analysis (maybe mention that you would evaluate the association between geographical location and the studied variables through the means of land use investigation maps).
Section 2.4:
- I believe it would be interesting to add descriptive statistics of the studied variables. The readers could benefit to see this kind of information to understand the data.
- It would be very instrutive to add a simple correlation matrix of the studied variables. It would help to describe/sumarize the associations and to detect some possible multicollinearity. If there is multicollinearity between the variables it would be advisable to remove redundant variables and, consequently, simplify the model.
- Also, please explain how you evaluated the fit/results of the CCA (mention or explain how you reported the results of the CCA mentioning the eingenvalues, Wilks lambda, etc).
Author Response
Response to Reviewer #1
We would like to thank the reviewer for careful and thorough reading of this manuscript and for the thoughtful comments and constructive suggestions, which help us to improve the quality of this manuscript. Our response follows.
General Comments:
The paper is well written and the question it tries to address is relevant. The original contribution of the study, as the authors augmented and I agree with them, is the use of large public datasets and spatial characteristics. This study tried to validate the results of previous studies about the same subject/question and I believe it managed to do so.
Reply:
We appreciate the positive feedback from the reviewer.
suggestions:
1. 14-15: “... and morbidity of diseases which considered as variables of human health in Taipei Metro.”
The sentence lacks something. Maybe the authors wanted to say: “... and morbidity of diseases which were considered as variables of human health in Taipei Metro.”?
Reply:
The suggested correction has been made as “… and morbidity of diseases which were considered variables of human health in Taipei Metro area.” (L. 14-15)
2. 35-36: Where it is written “to pursue more leisure and recreation, to contact with natural environment.”
Maybe rewrite to “to pursue more leisure and recreation, and to contact with natural environment.”
Reply:
The suggested correction has been made as “Amenities of green spaces encourage urban residents to increase physical activities, to pursue more leisure and recreation, and to contact with natural environment.” (L. 35-36)
3. 51-53: Where it is written: “cardiovascular disease (e.g. Pereira et al., 2012; Tamosiunas et al., 2014; Shen and Lung, 2016; Seo et al., 2019), type 2 diabetes (e.g. Astell-Burt et al., 2014; Bodicoat et al., 2014) .”
Maybe rewrite to “cardiovascular disease (e.g. Pereira et al., 2012; Tamosiunas et al., 2014; Shen and Lung, 2016; Seo et al., 2019), and type 2 diabetes (e.g. Astell-Burt et al., 2014; Bodicoat et al., 2014).”
Reply:
The suggested correction has been made. (L. 55)
4. 64-65: Where it is written: “There is only a new study extracted study population more than 300 thousand participants...”
Maybe rewrite the sentence because it is confusing.
Reply:
The sentence has been rewritten as “However, rare studies conducted their studies based on the large-scale datasets of human health insurance and analyzed the association between UGSs and the risk of disease in metropolitan areas (Seo et al., 2019).” (L66-70)
5. 88-89: Where it is written: “Therefore, canonical correlation analysis (CCA) was applied to determine what variables are more or less influential in the model.”
Maybe rewrite to: “Therefore, canonical correlation analysis (CCA) was applied to determine which variables are more or less influential in the model.”
Reply:
We have rewritten the sentence as “Therefore, canonical correlation analysis (CCA) was applied to determine which variables are more or less influential in the model.” (L. 92)
6. 111: The unit of measurement of rainfall is in meters? 2,400m in plain areas? Maybe convert to mm or in.
Reply:
The correction has been made. “The annual rainfall varies by terrain. The rainfall approximates 2,400mm in plain areas and is over 4,500mm in mountain areas (L. 114-115).”
7. 155-156: Where it is written: “which is a data subset of NHIRD, contains all original claim data for 1,000,000 beneficiaries randomly sampled...”
Maybe rewrite to: “which is a data subset of NHIRD, and contains all original claim data for 1,000,000 beneficiaries randomly sampled...”.
Reply:
The suggested correction has been made as “This study applied Longitudinal Health Insurance Database (LHID), which is a data subset of NHIRD, and contains all original claim data for 1,000,000 beneficiaries randomly sampled from the Registry for Beneficiaries (ID) of the NHIRD.” (L. 163-165)
8. Methods:
Section 2.1: This section lacks a reference for all the geographical information the authors provided about the Taipei Metro Area like rainfall, temperature, etc.
Reply:
Thank you for your comments.
The information about the number of districts and the total area of Taipei Metro area is derived from 2018 Statistical Yearbook of Ministry of the Interior (Ministry of the Interior, Republic of China (Taiwan), 2019). (L. 111-112)
Additionally, we have added an annotation to provide the reference about the information of rainfall and temperature data about the Taipei Metro Area. These information was collected from the 2018 Statistical Yearbook of Taipei City, New Taipei City, and Keelung City (Department of Budget, Accounting and Statistics, Taipei city government. 2019; Department of Budget, Accounting and Statistics, New Taipei City Government., 2019; Department of Budget, Accounting and Statistics, Keelung City government. 2019). (P. 3/17)
9. Section 2.2: Please, add a small paragraph explaining how you presented the results of the spatial analysis (maybe mention that you would evaluate the association between geographical location and the studied variables through the means of land use investigation maps).
Reply:
As suggested by the reviewer, we have added a small paragraph to explain how we presented the results of the spatial analysis.
“The landscape spatial characteristics of UGSs were analyzed for 37 districts of Taipei Metro Area. The results of the spatial analysis were used to examine the correlation between spatial characteristics of UGSs and morbidities of diseases in the study area.” (L. 148-150)
10. Section 2.4:
I believe it would be interesting to add descriptive statistics of the studied variables. The readers could benefit to see this kind of information to understand the data.
Reply:
As suggested by the reviewer, we have added Table 2 to show the descriptive statistics of the studied variables. It provides the information about the number of samples, minimum, maximum, mean, and standard deviation of the studied variables. (L. 211-213 and Table 2 in P. 6/17)
11. It would be very instructive to add a simple correlation matrix of the studied variables. It would help to describe/summarize the associations and to detect some possible multicollinearity. If there is multicollinearity between the variables it would be advisable to remove redundant variables and, consequently, simplify the model.
Reply:
As suggested by the reviewer, we have added a correlation matrix to describe the associations between the variables of spatial characteristics of UGSs and to detect the possibility of multicollinearity of studied variables. (L. 223-229)
Table 3 (L. 238) shows that all the correlation r < 0.765 (in absolute value). The general rule of thumb is that if the correlation > 0.8 then severe multicollinearity may be present. Therefore, it is not necessary to remove the redundant variables. (L. 226-228)
12. Also, please explain how you evaluated the fit/results of the CCA (mention or explain how you reported the results of the CCA mentioning the eingenvalues, Wilks lambda, etc.).
Reply:
As suggested by the reviewer, we have added a paragraph to explain how we evaluated the fit of the CCA and how we reported the results of the CCA.
“This study uses multivariate test of significance (i.e. Pillai’s, Hotelling’s, Wilk’s, and Roy’s multivariate criteria) to show the general fit of the model. Furthermore, the canonical correlation coefficients and the eigenvalues are used to explain the fitness of the canonical roots of the model.” (L. 229-232)
“Based on the tests the significance of each of the roots in the dimension reduction analysis of SPSS the possible significant roots are found. Since our model contains the three disease morbidities and six indicators of spatial characteristics of UGSs, SPSS extracts three canonical roots or dimensions. In the dimension reduction analysis, the first test of significance tests all three canonical roots of significance; the second test excludes the first root and tests roots two to three, the last test tests root three by itself.” (L. 232-237)
Reviewer 2 Report
Line 12. Insert “of” after “Most”.
Lines 14–15. Delete “as” [in “which considered as variables”].
Line 16. “source” not “sources”.
Line 52. Insert “and” before “type 2”.
Lines 64–65. “There is only a new study extracted study population”?? What does this mean? rewrite.
Line 91. Insert “an” before “accurate”.
Line 101. Insert “the” after “from”.
Line 111. “Mountain” doesn’t need to start with a capital letter.
Line 128. Forman and Godron 1986 is not in the References.
Line 131. Should “and” be inserted before “mean distance”? Something is wrong here and needs rewriting.
Line 136. “the” is needed before “Taipei”.
Line 163 and Line 164. Influenza does not need a capital “I”.
Line 170. Should read “summed up”. Delete “respectively” (it is unclear why this word is there).
Lines 189–190. The i’s and j’s need to be written as subscripts.
Line 205. Replace “correspondence” by “correlation”.
Line 216. Delete “the districts of”.
Lines 242–244. You neglect the second and third functions, but do not say what the basis was for this decision. Was the decision the result of the dimension reduction analysis (lines 246–252)? If so, then it should say so here. That is, present the results of the dimension reduction analysis first, and use it as the basis for restricting the interpretation to the first root. However, see the next comment.
Line 253. Table 3. Some researchers would argue that there is some significant information in the second root, as P=0.036 is commonly taken by statisticians to indicate statistical significance (i.e. 0.036 is less than 0.05). Whereas the first canonical variate reflects general health (since it involves all three criterion variables in almost equal measure), the second canonical variate is probably some kind of contrast between the diseases. It should be explored.
Line 261. Table 4. I question the value of including the column labelled Coef. (the standardized canonical function coefficient). It cannot be used by the readers and serves to confuse, because of the negative sign on Mental Disorders. As shown by the following column, 'mental disorders' contributes in the same direction as the other two criterion variables, i.e. low green spaces = high mental disorders. The important column is the following one, which contains the structure coefficient and its sign.
Lines 276–277. The low correlation of COHESION is shown in Table 4 and doesn’t need repeating. Don’t you really want to say something like “COHESION did not play a significant role in affecting human health”?
Line 283. Insert “the” before “three”.
Line 285. “positive” not “positively”.
Lines 286–296 & Figure 5. I would delete all of this material, as it contributes no information that is not already available in Tables 2 & 4. The canonical correlation coefficient of 0.79336 is in Table 2 and the structure coefficients of the predictor and criterion variables are in Table 4.
Lines 305–306. “has lower effects”? Do you mean “has a lesser effect”? This needs rewriting to better explain what you mean here.
Lines 316–317. “did not pay more attentions [should be ‘attention’] to other diseases”. Do you mean that it was concerned only with a single disease, viz. cardiovascular disease? If so, then you should say so directly.
Line 320–321. Insert “the” before “real”. Move “located” to the end of the sentence to read “where the patients were insured and outpatient clinics were located”.
Line 335. Replace “size” by “number”.
Line 337. Replace “the study” by “this study”.
Line 340. “and human health”. It is better to replace this by “and three diseases that humans suffer from” or something like that.
Lines 345–362. This is too long. Some of this material should be moved to the Discussion, e.g. the last paragraph (lines 357–362).
Line 375. Should read “References” not “Reference”.
Line 384. "australians" needs a capital 'A'.
Lines 385–386. No journal given.
Line 441. Reference 28. This should have only two authors I. Janssen & A.G. LeBlanc. The other authors, in which I. Janssen appears two more times, should be deleted.
Line 453, Reference 33. Delete this. It appears again below, in the correct alphabetical order.
Line 539. Delete “Social Science & Medicine”.
References. Lines 376–552. All the references need to be presented in the correct format and listed in the order in which they have been cited; that is not the case at present. There are many incorrect citations. Several journal titles have been cited as containing the word "Human" when it should be "Public", viz. Journal of Public Health, Annual Review of Public Health, International Journal of Environmental Research and Public Health. The style is not correct for presenting the author's surname and the abbreviation of the given name.
Author Response
Response to Reviewer #2
We would like to thank the reviewer for careful and thorough reading of this manuscript and for the thoughtful comments and constructive suggestions, which help us to improve the quality of this manuscript. Our response follows.
1. Line 12. Insert “of” after “Most”.
Reply:
The correction has been made. (L. 12)
2. Lines 14–15. Delete “as” [in “which considered as variables”].
Reply:
The correction has been made. (L. 14-15)
3. Line 16. “source” not “sources”.
Reply:
The correction has been made. (L. 17)
4. Line 12. Insert “of” after “Most”.
Reply:
The correction has been made. (L. 12)
5. Line 52. Insert “and” before “type 2”.
Reply:
We have inserted “and” before type2 …. (L. 55)
6. Lines 64–65. “There is only a new study extracted study population”?? What does this mean? rewrite.
Reply:
As suggested by the reviewer, we have rewritten the sentence.
“However, rare studies conducted their studies based on the large-scale datasets of human health insurance and analyzed the association between UGSs and the risk of disease in metropolitan areas (Seo et al., 2019).” (L. 66-70).
7. Line 91. Insert “an” before “accurate”.
Reply:
The correction has been made. (L. 94)
8. Line 101. Insert “the” after “from”.
Reply:
We have inserted “the” after “from”. (L. 104)
9. Line 111. “Mountain” doesn’t need to start with a capital letter.
Reply:
The correction has been made. (L. 115)
10. Line 128. Forman and Godron 1986 is not in the References.
Reply:
We have added the reference of Forman and Godron (1986).
“24. Forman, R.T.T.; Godron, M. Landscape Ecology. Wiley, New York, 1986.” (L. 517)
11. Line 131. Should “and” be inserted before “mean distance”? Something is wrong here and needs rewriting.
Reply:
We have rewritten the sentences.
“The proportion of green spaces (PLAND) was calculated to assess the availability of UGSs in the Taipei Metro area. This study also selected shape of UGS (SHAPE_MN), mean distance of UGS patches to the nearest residential patch (ENN_MN), and continuity of UGS patches (COHESION) as the indicators of spatial pattern of UGSs.” (L. 133-137)
12. Line 136. “the” is needed before “Taipei”.
Reply:
The correction has been made (L. 140).
13. Line 163 and Line 164. Influenza does not need a capital “I”.
Reply:
The correction has been made (L. 171).
14. Line 170. Should read “summed up”. Delete “respectively” (it is unclear why this word is there).
Reply:
The correction has been made (L. 178).
15. Lines 189–190. The i’s and j’s need to be written as subscripts.
Reply:
The correction has been made.
“The goal of CCA is to determine the coefficients, or canonical weights (aij and bij), that maximize the correlation between canonical variates Ui and Vi. The first canonical correlation, Corr. (U1, V1), is the strongest possible correlation between a linear combination of variables in the exposure set and a linear combination of variables in the outcome set.”
16. Line 205. Replace “correspondence” by “correlation”.
Reply:
We have replaced “correspondence” by “correlation” (L. 216).
17. Line 216. Delete “the districts of”.
Reply:
The correction has been made (L. 250).
17. Lines 242–244. You neglect the second and third functions, but do not say what the basis was for this decision. Was the decision the result of the dimension reduction analysis (lines 246–252)? If so, then it should say so here. That is, present the results of the dimension reduction analysis first, and use it as the basis for restricting the interpretation to the first root. However, see the next comment.
Line 253. Table 3. Some researchers would argue that there is some significant information in the second root, as P=0.036 is commonly taken by statisticians to indicate statistical significance (i.e. 0.036 is less than 0.05). Whereas the first canonical variate reflects general health (since it involves all three criterion variables in almost equal measure), the second canonical variate is probably some kind of contrast between the diseases. It should be explored.
Reply:
As suggested by reviewer, we have rewritten the paragraph about the output of dimension reduction analysis. We neglect only the third functions based on the tests the significance of each of the roots. We include the Function 2 in the table 7 to shows the statistics of the canonical solution of the relationships between spatial characteristics of UGSs and morbidities of diseases. We have also added a paragraph to explain the canonical correlation solution of Function2.
“SPSS extracts three canonical roots or dimensions for our model. The dimension reduction analysis tests the significance of each of the roots (see Table 6). The first test of significance shows the full model across root 1 to 3 is significant (Wilk’s λ = 0.19831, F = 3.41680, p < 0.01). The second test excludes the first root and tests roots 2 to 3 and shows it also is significant (Wilk’s λ = 0.53512, F = 2.21872, p = .036 < 0.05). The root 3 itself is not significant (Wilk’s λ = 0.90290, F = .80659, p = 0.531) (see Table 6).” (L. 290-300)
“In the Function 2, the values of structure coefficients (rs) indicate that the variables of PLAND, NDVI and COHESION are the primary contributors to the synthetic variate of predictors. The cross-structure coefficients (rc) of PLAND, NDVI, and COHESION show that they are positively related to the synthetic variate of disease morbidities with moderate rc = 0.34186, 0.33197, and 0.47943. On the other side of the Function 2, the structure coefficients (rc) of criterion variables indicate that cardiovascular diseases, mental disorders, and respiratory diseases have very low correlation coefficients with the synthetic variate of morbidity of diseases (with very low rs < 0.2). The cross-structure coefficients (rc) of all the three criterion variables are lower than 0.12, reveal that morbidity of cardiovascular diseases, mental disorders, and respiratory diseases all have very low correlations with the synthetic variate of spatial characteristics of UGSs.” (L. 326-335)
18. Line 261. Table 4. I question the value of including the column labelled Coef. (the standardized canonical function coefficient). It cannot be used by the readers and serves to confuse, because of the negative sign on Mental Disorders. As shown by the following column, 'mental disorders' contributes in the same direction as the other two criterion variables, i.e. low green spaces = high mental disorders. The important column is the following one, which contains the structure coefficient and its sign.
Reply:
We do agree the comment. We have removed the column labelled Coef. (the standardized canonical function coefficient). The new table (table 7) shows the statistics of canonical solution of Function 1 and Function 2 and presents the structure coefficients (rs), the squared structure coefficients (rs2), and cross-structure coefficients (rc) between a set of variables and the synthetic variable created by another set of variables. (L. 360-363)
19. Lines 276–277. The low correlation of COHESION is shown in Table 4 and doesn’t need repeating. Don’t you really want to say something like “COHESION did not play a significant role in affecting human health”?
Reply:
As suggested by reviewer, we have rewritten the paragraph to clarify the explain the new table (table 7) about the canonical correlation solution of Function 1.
“In the Function 1, the first pair of canonical variates groups the variables in the way that the correlation between them is maximized. The structure coefficients (rs) of PLAND, NDVI, SHAPE_MN, ENN_MN and Network Distance indicate that they are the primary contributors (with rs = -0.53151, -0.60826, -0.48613, 0.53458, and 0.88403) to the synthetic variate of characteristics of green spaces. The cross-structure coefficients (rc) of PLAND, NDVI, and SHAPE_MN show that they are negatively related to the synthetic variate of morbidity of diseases. Inversely, the cross-structure coefficients (rc) for ENN_MN and Network Distance show that they are positively related to the synthetic variate of morbidity of diseases. The other side of the Function 1, the structure coefficients (rs) of criterion variables indicate that cardiovascular diseases, mental disorders, and respiratory diseases are positively correlated with the synthetic variate of morbidity (with very high rs = .98556, .98352, and .99449). The cross-structure coefficients (rc) of the three criterion variables, all of which are higher than 0.7, reveal that morbidity of cardiovascular diseases, mental disorders, and respiratory diseases all have positive correlations with the synthetic variate of spatial characteristics of UGSs.” (L. 312-325)
20. Line 283. Insert “the” before “three”.
Reply:
We have inserted “the” before “three” (L. 322).
21. Line 285. “positive” not “positively”.
Reply:
The correction has been made (L. 324).
22. Lines 286–296 & Figure 5. I would delete all of this material, as it contributes no information that is not already available in Tables 2 & 4. The canonical correlation coefficient of 0.79336 is in Table 2 and the structure coefficients of the predictor and criterion variables are in Table 4.
Reply:
As suggested by reviewer, we have deleted the Figure 5 and also removed the interpretations about Figure 5 from section 3.
23. Lines 305–306. “has lower effects”? Do you mean “has a lesser effect”? This needs rewriting to better explain what you mean here.
Reply:
As suggested by reviewer, we have written the sentence.
“The availability of UGSs (measured by PLAND) “is less influential” in the relationship between spatial characteristics of UGSs and human health.”
24. Lines 316–317. “did not pay more attentions [should be ‘attention’] to other diseases”. Do you mean that it was concerned only with a single disease, viz. cardiovascular disease? If so, then you should say so directly.
Reply:
As suggested by reviewer, we have rewritten two sentences in the paragraph.
“However, the previous study usually concerned only with a single disease, i.e. cardiovascular disease. Our study reveals the correlations between spatial characteristics of UGSs and morbidity of three diseases.” (L. 398-401)
25. Line 320–321. Insert “the” before “real”. Move “located” to the end of the sentence to read “where the patients were insured and outpatient clinics were located”.
Reply:
The correction has been made.
“A major limitation of our study is that the data of National Health Insurance Research do not reveal the real addresses of the patients but only the districts where the patients were insured and outpatient clinics located.” (L. 338-340)
26. Line 335. Replace “size” by “number”.
Reply:
The correction has been made. (L. 419)
27. Line 337. Replace “the study” by “this study”.
Reply:
The correction has been made. (L. 421)
28. Line 340. “and human health”. It is better to replace this by “and three diseases that humans suffer from” or something like that.
Reply:
As suggested by reviewer, we have written the sentence.
“This study shows the detailed relationships between spatial characteristics of UGSs and three diseases that humans suffer from based on the data of morbidities of three diseases.” (L. 421-423)
30. Lines 345–362. This is too long. Some of this material should be moved to the Discussion, e.g. the last paragraph (lines 357–362).
Reply:
As suggested by reviewer, we have move the first sentence away. Moreover, we have deleted the second sentence (L. 422-426).
31. Line 375. Should read “References” not “Reference”.
Reply:
The correction has been made. (L. 462)
32. Line 384. "australians" needs a capital 'A'.
Reply:
The correction has been made. (L. 470-470)
33. Lines 385–386. No journal given.
Reply:
The reference has been corrected.
“5. Avron, H.; Boutsidis, C., Toledo, S., & Zouzias, A. Efficient dimensionality reduction for canonical correlation analysis. In: International Conference on Machine Learning 2013 (pp. 347-355).” (L. 472-473)
34. Line 441. Reference 28. This should have only two authors I. Janssen & A.G. Le Blanc. The other authors, in which I. Janssen appears two more times, should be deleted.
Reply:
The reference has been corrected.
“32. Janssen, I.; Le Blanc, A. G. Systematic review of the health benefits of physical activity and fitness in school-aged children and youth. International journal of behavioral nutrition and physical activity 2010, 7(1), 40.” (L. 535-536)
35. Line 453, Reference 33. Delete this. It appears again below, in the correct alphabetical order.
Reply:
The redundant reference has been deleted.
36. Line 539. Delete “Social Science & Medicine”.
Reply:
The correction has been made.
“70. De Vries, S.; Van Dillen, S. M.; Groenewegen, P. P.; Spreeuwenberg, P. Streetscape greenery and health: stress, social cohesion and physical activity as mediators. Social science and medicine 2013, 94, 26-33.” (L. 626-627)
37. Lines 376–552. All the references need to be presented in the correct format and listed in the order in which they have been cited; that is not the case at present. There are many incorrect citations. Several journal titles have been cited as containing the word "Human" when it should be "Public", viz. Journal of Public Health, Annual Review of Public Health, International Journal of Environmental Research and Public Health. The style is not correct for presenting the author's surname and the abbreviation of the given name.
Reply:
Thanks a lot for the reviewer’s careful and thorough reading of this manuscript. As you pointed out that our references in last version really needed to be improved. We have already checked all the references carefully and made sure that all the citations are correct. We have already replaced “human health” by “public health” of Journal titles to corrected the journal title. Please see the new “References”. (L. 462-637)